# Grade Setting of a Timber Logistics Center Based on a Complex Network: A Case Study of 47 Timber Trading Markets in China

**Liang Xue [1] , Xin Huang [2,*], Yuchun Wu [3], Xingchen Yan [1] and Yan Zheng [1]**

1   College of Automobile and Traffic Engineering, Nanjing Forestry University, Longpan Road 159#, Nanjing 210037, China; shiling97322@163.com (L.X.); xingchenyan.acad@gmail.com (X.Y.); yzheng_x@163.com (Y.Z.)
2   College of Civil Engineering, Nanjing Forestry University, Longpan Road 159#, Nanjing 210037, China
3   Collaborative Innovation Center of steel Technology, University of Science and Technology Beijing, Xueyuan Road 30#, Beijing 100083, China; wuyuchun@njfu.edu.cn
*   Correspondence: huangxin@njfu.com.cn; Tel.: +86-139-5197-0539

**Abstract:** The location and grade setting of a timber logistics center is an important link in the optimization of the timber logistics system, the rationality of which can effectively improve the efficiency of the timber logistics supply chain. There is a long distance between the main forested areas in China, and more than 55% of the timber demand depends on imports. Research and practice of systematically planning timber logistics centers in the whole country have not been well carried out, which reduces the efficiency of timber logistics. In this paper, 47 timber trading markets with a certain scale in China are selected as the basis for logistics center selection. Based on their transportation network relationship and the number of enterprises in the market, combined with the complex network theory and data analysis method, the network characteristics of three different transportation networks are measured. After determining the transportation capacity indicator, the logistics capacity coefficient is measured based on the freight volume of each node. Then, the important nodes are identified, and each node is graded to systematically set up the timber logistics center.

**Keywords:** complex network; data analysis; timber; logistics center; grade setting

---

## 1. Introduction

The nodes in the supply chain of timber products include forestry enterprises, timber processing enterprises, furniture enterprises, timber distribution enterprises, etc. For all participants in the supply chain, the efficiency and cost of the logistics system are the key factors that affect the efficiency of the enterprise. As important parts of the whole timber logistics system, reasonable location selection and grade setting can effectively promote the optimal allocation of timber flow in the process of timber production and consumption, are directly related to the cost and expense level of logistics businesses and, more importantly, can provide customers with high-quality and efficient logistics services. In view of the distribution of timber resources and customers, the supply and demand of timber, the transportation network, land price level and natural conditions, the distribution of import and export ports and other factors which lead to the establishment of timber logistics centers in different places, the economic benefits of the whole timber logistics system and the impact of the operation process of the system on the environment are different. At present, there is a lack of systematic and standard timber logistics center planning and grade setting, so it is necessary to study this topic.

Scholars used the method of combining qualitative and quantitative analyses for choosing a logistics center. Sun et al. (2008) [1] proposed a bilevel programming model based on a heuristic

algorithm. The upper-level model is to determine the optimal location by minimizing the planners' cost, and the lower gives an equilibrium demand distribution by minimizing the customers' cost, so as to find the best location for a logistics distribution center. Marković et al. (2013) [2] considered the significance of selecting and ranking different locations. It is necessary to compare, as objectively as possible, the influences of various criteria and reduce them to a common function, i.e., present the methodology for solving complex problems associated with the ranking of alternatives. Önden et al. (2016) [3] determined that 19 logistics centers in Turkey are developing a systematic approach and integrating different transport modes to improve logistics performance. To reach suitability levels, seven decision criteria were considered with their priority levels. Zhang et al. (2017) [4] proposed a rural logistics center location model based on the theory of intuitionistic fuzzy TOPSIS. On the basis of the evaluation indicator system, the information is integrated according to the experts' score. The entropy weight method is used to determine the weight of each evaluation indicator. The results are sorted by intuitionistic fuzzy TOPSIS. Vanessa et al. (2017) [5] noted that the logistics integration center (LIC) has become an attractive option to realize efficient intermodal transport. The mixed integer linear programming model is applied to the soybean flow in Brazil, which can be used to evaluate whether the transportation cost will be reduced if there are LICs. Zhao (2018) [6] noted that when deciding medicine logistics center location with financial results in mind, market size, growth and demand creation must be the primary considerations. Mousavi et al. (2019) [7] presented a new MCGDM model that applied the concept of a compromise solution under uncertainty. Then, a new Collective Indicator was developed that simultaneously regards the distances of cross-docking centers as candidates from the IVIF-ideal points.

There has been a certain research focus on the use of complex network theory to study the importance of transportation network nodes and network characteristics. Xia et al. (2010) [8] used an iterative algorithm to optimize the allocation of node capacity considering the uneven transportation generation rate of different nodes. Cardillo et al. (2013) [9] used real data from the European Air Transport Composite Network to analyze the structural characteristics of an essentially multilayer real network. Couto et al. (2015) [10] analyzed the complex network characteristics of the Brazilian air transport network and confirmed that the network had small-world characteristics, while the national connection network followed a power law distribution. Sun et al. (2016) [11] found that most of Dublin's important nodes and transportation hubs were located in the city center, i.e., most bus lines often intersect with the city center. Ducruet et al. (2017) [12] confirmed that the development of the network is mainly concentrated around large hubs and that the transportation distribution depends on the location of important nodes that have been established. Yu et al. (2019) [13] focused on the change of space structure of the Nanjing Metro network combined with the concept of node degree in a complex network to conclude that line connection tends to occur at important nodes. Zhang et al. (2019) [14] used the complex network method to characterize freight transport, which showed that the city-scale network at the prefecture and county levels had obvious small-world network characteristics.

Research on the grade setting and evaluation of a logistics center is not enough. Shi et al. (2011) [15] established the evaluation indicator system of a logistics green level for a coal distribution center. By using the fuzzy analytic hierarchy process (AHP) evaluation method, the evaluation vector of an alternative power plant and port logistics green level was obtained, which is decisive for the final site selection. Yang et al. (2015) [16] proposed the support vector machine, which can promote generalization through structural risk minimization, thus solving practical problems, such as small sample data sets, nonlinearity, high dimensions and local minima, which play significant roles in evaluating the ability of a military logistics distribution center. Wu (2016) [17] proposed a multiattribute group decision-making (MAGDM) framework to facilitate such evaluations. The MAGDM framework was applied to the lean practice evaluation of the logistics distribution center of a commercial tobacco company. Chen (2017) [18] described the function of an urban logistics center and put forward the location principle and construction model for one. The final evaluation indicator system was composed of six first-level indicators and four second-level indicators. Liu et al. (2019) [19] proposed a method for

selecting the location of a pharmaceutical logistics center based on combined weight and cumulative prospect theory. The comprehensive evaluation model of pharmaceutical logistics was established, which solves the comprehensive prospect value of each pharmaceutical logistics center and determines the competitiveness sequence. Li et al. (2019) [20] proposed the use of AHP to select four indicators as the evaluation and judgment basis to evaluate the operation system of the logistics distribution center of steel enterprises.

Scholars have performed relatively little research on timber logistics transportation and related networks. Lin et al. (2016) [21] used an improved K-means clustering analysis algorithm and achieved recognition of vehicle type and timber transportation volume in the process of timber transportation using neural network technology. Long et al. (2016) [22] constructed an international trade network of woody forest products with countries as nodes and trade relations as links based on the complex network theory and analyzed the characteristics of the network. Lin et al. (2017) [23] analyzed and designed a timber logistics network system using Web GIS technology, database technology and component integration technology. Sarrazin et al. (2019) [24] proposed a profit maximization model that considers the interaction between forest logistics centers and complex forest networks and reduces sorting and transportation costs. Long et al. (2019) [25] used global timber and forest products trade data from 2004 to 2016 to construct a global unweighted and weighted forest products competition network from the perspective of import and introduced five indicators to study its spatial distribution.

Until now, scholars have mainly focused on studying the logistics center location based on the quantitative method supplemented by the qualitative method. Most quantitative studies have considered the use of complex mathematical models and obtained the optimal solution for a logistics center. However, the calculation results are often unsatisfactory because they involve the construction cost of the logistics center, the surrounding transportation, the economic benefits and other factors. In addition, the analytic hierarchy process (AHP), expert scoring and fuzzy analysis was used for qualitative research; some views are relatively subjective, and were also slightly inadequate. There are many studies that combine the two, but the final results may change due to the actual environment and conditions, which will make the research results have no practical significance.

The development of timber logistics is limited by the characteristics of timber. Compared with other goods, timber occupies more space and has irregular shapes, which leads to problems in stacking. In regard to the choice of the transportation mode, it is not as random as that used for other goods. General timber transportation uses highway transportation, followed by railway transportation, and air transportation is used relatively less frequently. The construction of a timber logistics center is the carrier of timber logistics and is also an important way to develop regional logistics. Therefore, it is necessary to fully consider the differences in the functions of regional logistics centers when building timber logistics centers in various regions to correctly set the grade of the logistics centers.

This paper mainly studies the characteristics of the transportation network, which is composed of the timber trading markets. On this basis, a reasonable method for determining the location and grade setting of the timber logistics center is proposed, which is also based on the transportation capacity and freight volume indicator. The research object is an alternative logistics center composed of 47 timber trading markets with a certain scale in China.

## 2. Methods

### 2.1. Basic Types of Networks

Networks can basically be divided into unweighted networks and weighted networks. The difference between the two types of networks is that the definition of the relationship between nodes is inconsistent, and the network uses the lines between nodes to express the relationship between them. The unweighted network divides the relationship between nodes into "yes" and "no", corresponding to the numbers "1" and "0", respectively, while the weighted network weights the distance between nodes as the relationship value.

The unweighted network is generally expressed as $G(V, E)$, where the $V$ set represents the node, the node represents the distribution node in the distribution network and $E$ represents the relationship set between the nodes. The weighted network is generally expressed as $F(V, W)$; the element $w_{ij}$ in the relational matrix represents the weight added between nodes $v_i$ and $v_j$, and $w_{ij}$ is greater than or equal to 0.

*2.2. Basic Characteristics of Networks*

Complex networks are abstract representations of real systems. To further analyze and optimize networks, it is necessary to measure and analyze their basic characteristics. The network studied in this paper is an undirected graph.

(1) Centrality

Centrality is the most basic indicator, which is generally recorded as $K_i$ and includes relative centrality and absolute centrality. The centrality of node $P$ is the number of other nodes directly connected with node $P$. The higher the centrality of a node, the more edges the node has with many nodes. The higher the centrality of a node in the transportation network, the better the transportation accessibility of that node in the network.

(2) Betweenness

Complex networks have betweenness of edges and nodes. The edge betweenness of edge $e_i$ refers to the number of shortest paths passing through the edge in a network. The betweenness of node $v_i$ refers to the number of shortest paths passing through node $i$ in the network. If a certain node in the transportation network has a larger node number, then the hub type and betweenness of the node in the network are better. Similarly, the edge-to-edge number can reflect the importance of edges in the network and help distinguish between the advantages and disadvantages of paths. The edge-to-edge number is generally expressed by $B_i$.

(3) Node strength

Node strength is the concept of betweenness in weighted networks. The node strength of node $P$ is the sum of the weights between other nodes directly connected with node $P$. The number of nodes adjacent to the node and the weights between adjacent nodes are considered, reflecting the comprehensive connection of the node. Generally, node strength is expressed as $S_i$, and the connection of weighted networks is recorded. The join matrix is $W$, $N_i$ denotes the node matrix edged with node $i$, and the formula for node strength is as follows.

$$S_i = \sum_{j \in N_i} W_{ij} \tag{1}$$

(4) Unit weight

Based on node strength and betweenness, the concept of unit weight is introduced, which is denoted as $U_i$, and is the ratio of node strength to betweenness. The formula used to reflect the comprehensive information of nodes and their adjacent nodes and to describe the connection relationship and weight of nodes in more detail is shown below. The unit weight indicator reflects the distance coefficient between nodes and other nodes in the transportation network. The larger the unit weight, the better the transportation performance of the nodes.

$$U_i = \frac{S_i}{K_i} \tag{2}$$

(5) Node difference

To distinguish between nodes with the same betweenness, node strength and unit weight, the concept of edge weight difference is proposed. By measuring the weights of each edge in the network, we can determine whether the neighboring edges of the node have the same edge weights or whether only a few edges have larger edge weights. Generally expressed as $Y_i$, the formula is as follows. In the formula, $N_i$ is combined with nodes with edge connections with node *i*. The indicator reflects the difference of the weights between nodes. The greater the difference of nodes, the better the transportation accessibility.

$$Y_i = \sum_{j \in N_i} (\frac{W_{ij}}{S_i})^2 \tag{3}$$

*2.3. Analysis Method for the Timber Logistics Transportation Network Based on a Complex Network*

Problem Description

The timber logistics transportation network includes the routes between transportation nodes and nodes. The following assumptions are proposed before establishing the model:

(1) The freight volume and transportation capacity of nodes are the main factors that affect the accessibility of the logistics transportation network.
(2) The distance, route and time between nodes are based on the existing transportation network.
(3) Intercity freight transportation mainly adopts the multimodal transportation of air and railway transportation, while highway transportation is used in the same city.
(4) The freight capacity of each node (market) settled enterprise is equivalent, so the number of enterprises in the market is used as the indicator for measuring its freight capacity.
(5) The direction of transportation is not considered in the course of transportation.

*2.4. Transportation Network Models*

The model in this paper has no objective or constraint conditions. The model is only a problem description and analysis method that uses a complex network to measure the characteristics of the network and express its transportation accessibility and freight demand.

2.4.1. Data Acquisition

Data mining technology in the era of big data is often based on computer frontier theories, such as machine learning and artificial intelligence. Common methods include classification models, regression analysis, clustering analysis, association rule analysis and web data mining.

Since the transportation analysis data in this paper include air, railway and highway transportation relationships among nodes, the relationship data, mileage and transportation time exist on many websites, including China Railway 12306, Variflight route map and others. To obtain data between nodes quickly, some of the data in this paper are obtained using the Python multiprocess crawler program.

The data in this paper are based on 47 nodes in three modes of transportation, namely, air, railway and highway transportation; their weighted and unweighted relation matrix; and the freight demand of each node. The initial data settings contains six 47 by 47 matrices, and combined with UCINET analysis, four transportation capacity indicators, freight demand indicators and the logistics capacity coefficients under three modes of transportation are obtained.

2.4.2. Complex Transportation Network Model

Based on the above assumptions, quantitative indicators should be introduced to analyze the transportation capacity, namely, the transportation accessibility between nodes, node centrality, unit distance between nodes and distance difference between nodes. These four indicators

are determined by the degree of nodes, betweenness, unit weight and difference of nodes in complex networks.

The edge-connection relationship of nodes is transformed into an adjacency matrix, which is imported to Ucinet. The node degree and betweenness centrality for nodes of unweighted networks are measured with MATLAB. The transportation accessibility and node centrality of each node are calculated to provide data support for the measurement of weighted networks.

There is a long distance between the markets using air and railway transportation analysis, which is across provinces and cities; the transportation distance is directly proportional to the transportation cost. It is feasible to use the transportation distance and time as the relation matrix. However, the charging rules are simple between nodes in the same city, so it is easier to understand that the transportation cost is directly proportional to the transportation time.

In addition, to unify the indicator, the value of $W_{ij}$ adopts a similar weight, i.e., the larger the value of $W_{ij}$, the closer the relationship between nodes. For the distance and time of transportation, reciprocal processing is carried out to obtain the *F(V, W)* matrix. Note that since there is not an obvious difference in speed between air transportation and highway transportation, the transportation distance is chosen as the original weight. The choice of railway transportation is diverse, and the speeds of ordinary trains and high-speed trains are quite different, so the transportation time is chosen as the original weight.

After processing, the relationship matrix is imported into Ucinet and combined with the characteristics of the unweighted network measurement. Matrix processing software is used to measure the indicator of the unit distance between nodes and the difference of the distance between nodes.

### 2.4.3. Freight Volume Analysis and Indicator Processing

Considering that the freight volume of each node is also an important factor that affects the logistics performance of the node, the freight volume is described by the number of enterprises in each market. Thus far, we can obtain the freight volume indicator and four transportation capacity indicators; namely, the transportation accessibility between nodes, the betweenness of nodes, the unit distance between nodes and the difference between nodes are obtained.

Since the characteristic value units of measurement are different, which is not conducive to direct operation, the data are required to be dimensionless. The specific indicator processing methods are as follows:

(1)　The four transportation indicators and freight volume indicators measured by Ucinet are processed by the Z-score to make them dimensionless.

(2)　The four indicators under the three modes of transportation are summed to obtain the corresponding transportation capacity value. Then, the comprehensive transportation capacity of each node is calculated by weighting the transportation capacities of railway and air and adding the transportation capacity of highway. Combined with the freight volume indicator, the logistics capacity coefficient of each node is obtained.

(3)　The city is selected as the unit to formulate the rules of network construction. The first 10% of cities are scored as the highest level, the second 20% of cities are scored as the second level, the next 30% of cities are scored as the third level and the last 40% of cities are scored as the fourth level. The node with the highest score is selected as the logistics center in each city.

## 3. Empirical Study of the Complex Structural Characteristics of the Timber Transportation Network

### 3.1. Problem Description

From the "China Timber Network", 47 timber markets nationwide are used to build a timber logistics transportation network, which includes a total of 10 provinces, 19 prefectural-level cities and

3 municipalities. Among them, Guangzhou has the largest number of timber markets, at 20. The timber market numbers and details are shown in Table 1 below.

**Table 1.** Details of 47 timber markets.

| Node Number | Timber Market Name |
| --- | --- |
| 1 | Dongguan Houjie·Xingye Timber Market |
| 2 | Dongguan Jiaye Timber Market |
| 3 | Dongguan Houjie·Oriental Xingye City |
| 4 | Guangdong Dongguan Jilong Timber Market |
| 5 | Dongguan Chashan Fengye Timber Market |
| 6 | Foshan Shunde ShuiTeng Timber Market |
| 7 | Foshan Big Turn Splint Market |
| 8 | Foshan Shunde Asia Pacific Timber Market |
| 9 | Foshan Shunde Longshan Decorative Materials Market |
| 10 | Guangdong Foshan Nanhai Bomei Market |
| 11 | Guangzhou Xintang Yueshun Decoration Materials City |
| 12 | Guangzhou Tianma Decoration Materials Market |
| 13 | Guangzhou Panyu Wuzhou Decoration World |
| 14 | Guangdong Guangzhou Yuzhu International Timber Market |
| 15 | Guangzhou Yangcheng Decorative Materials Market |
| 16 | Guangzhou Xilong Building Materials Market |
| 17 | Guangzhou B&Q Building Materials Supermarket |
| 18 | Guangzhou Tianjian Decoration Materials Market |
| 19 | Guangzhou Anhua Decorative Materials Market |
| 20 | Guangzhou Yiheng Timber Market |
| 21 | Hangzhou Timber Trading Market |
| 22 | Shaoxing Tashan Timber Market |
| 23 | Wenzhou Timber Exchange Market |
| 24 | Jiaxing Jiashan International Timber Market |
| 25 | Jiaxing Jiashan·Oriental Xingye City |
| 26 | Huzhou Nanxun Building Materials Market |
| 27 | Huzhou Timber Market |
| 28 | Zhangjiagang Timber Exchange Market |
| 29 | Taicang City Port District Huachen Logistics Center |
| 30 | Zhenjiang Yangzhong Venus Timber Market |
| 31 | Tianjin Xinzhong Building Materials Trade Market |
| 32 | Tianjin Bohai Bay Timber Market |
| 33 | Tianjin Beichen District Hanjiashu Timber Market |
| 34 | Shandong Dezhou Timber Market |
| 35 | Qingdao Jiaozhou Timber Market |
| 36 | Qingdao Licang District Wantou First Timber Market |
| 37 | Hunan Changsha Zhongnan Timber Market |
| 38 | Yueyang City Sun Bridge Building Materials Market |
| 39 | Shijiazhuang Zhengding Hengshan Plate Market |
| 40 | Shijiazhuang City Timber Trading Market |
| 41 | Shanghai Jiading District Donghua Global Building Materials Market |
| 42 | Shanghai Baoshan Furen Timber Market |
| 43 | Beijing Dongba Imported Timber Market |
| 44 | Heilongjiang Suifenhe Imported Timber market |
| 45 | Inner Mongolia Manzhouli City Timber Market |
| 46 | Anhui Jieshou Lvzhai Timber Market |
| 47 | Jiangxi Jinxian Wengang Wild Chicken Cage Timber Market |

The transportation mode varies according to the timber product type. Based on the transportation network constructed using the 47 timber markets, this paper chooses three modes of transportation to analyze the transportation capacity of each node: highway, railway and air.

### 3.2. Analysis of the Complex Transportation Network Model

3.2.1. Analysis of the Air Transportation Network

Since there are a small number of rare timber products, including ebony, aloes, rosewood, yellow pear wood, etc., the safety of timber in the transportation process must be ensured. Therefore, air transportation is generally used. According to the specific location of each timber market and the scope of the city, it is necessary to determine whether there are air transportation resources. According to statistics, the air transportation conditions of the timber market are shown in Table 2 below (enumeration):

**Table 2.** Air transportation conditions in 47 timber markets (enumeration).

| Node Number | City Name | Air Transportation Conditions | Node Number | City Name | Air Transportation Conditions |
|---|---|---|---|---|---|
| 6 | Foshan | Foshan Shadi Airport | 30 | Zhenjiang | No Airport |
| 7 | Foshan | Foshan Shadi Airport | 31 | Tianjin | Tianjin Binhai Airport |
| 10 | Foshan | Foshan Shadi Airport | 34 | Dezhou | No Airport |
| 11 | Guangzhou | Guangzhou Baiyun Airport | 35 | Qingdao | Qingdao Liuting Airport |
| 13 | Guangzhou | Guangzhou Baiyun Airport | 37 | Changsha | Changsha Huanghua Airport |
| 14 | Guangzhou | Guangzhou Baiyun Airport | 38 | Yueyang | Yueyang Sanhe Airport |
| 16 | Guangzhou | Guangzhou Baiyun Airport | 40 | Shijiazhuang | Shijiazhuang Zhengding Airport |
| 17 | Guangzhou | Guangzhou Baiyun Airport | 41 | Shanghai | Shanghai Hongqiao Airport |
| 19 | Guangzhou | Guangzhou Baiyun Airport | 43 | Beijing | Beijing Capital Airport |
| 21 | Hangzhou | Hangzhou Xiaoshan Airport | 45 | Hulun Buir | Western suburbs of Manchuria |
| 22 | Shaoxing | No Airport | 46 | Fuyang | Fuyang Xiguan Airport |
| 23 | Wenzhou | Wenzhou Longwan Airport | 47 | Nanchang | Nanchang Changbei Airport |

(1) Analysis of the Air Transportation Network

Based on domestic flights, the relationship between airports is transformed into an adjacent matrix $G_a$, which is analyzed using Ucinet. The matrix diagram is shown in Figure 1. From the diagram, the timber markets in Dongguan and Jiaxing are not connected with other markets in terms of air transportation, which is because there is no air freight transportation condition in the cities where these timber markets are located. However, Shanghai, Wenzhou, Tianjin, Qingdao and other timber markets are located in the middle of the figure, which shows that there are many edge connections between these markets and other markets in terms of air freight transportation.

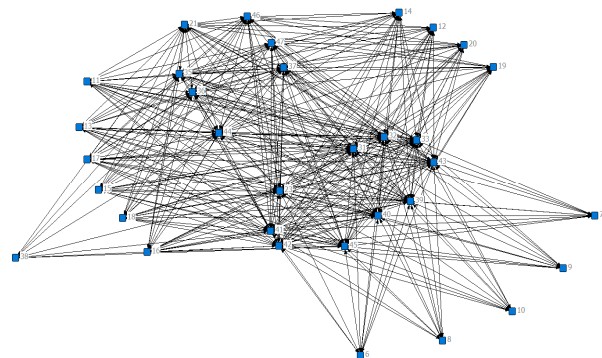

**Figure 1.** Unweighted network diagram of air transportation.

Through centrality and weighted measurement analysis, the node degree and betweenness centrality for each node of the air transportation network are determined. The characteristics are detailed in Table 3 (with the top 10 listed). In the table, Degree refers to the node degree of each node, nDegree refers to the relative node degree, Betweenness refers to the betweenness centrality of the node and nBetweenness refers to the relative betweenness centrality of the node.

**Table 3.** Node degree and betweenness centrality of the air transportation unweighted network.

| Node Number | Degree ($K_i$) | nDegree | Betweenness ($B_i$) | nBetweenness |
|---|---|---|---|---|
| 1 | 0.000 | 0.000 | 0 | 0 |
| 2 | 0.000 | 0.000 | 0 | 0 |
| 3 | 0.000 | 0.000 | 0 | 0 |
| 4 | 0.000 | 0.000 | 0 | 0 |
| 5 | 0.000 | 0.000 | 0 | 0 |
| 6 | 10.000 | 0.217 | 0.456 | 0.044 |
| 7 | 10.000 | 0.217 | 0.456 | 0.044 |
| 8 | 10.000 | 0.217 | 0.456 | 0.044 |
| 9 | 10.000 | 0.217 | 0.456 | 0.044 |
| 10 | 10.000 | 0.217 | 0.456 | 0.044 |

(2) Analysis of Air transportation Distance Analysis

To ensure the indicator changes in the same direction, the reciprocal distance is taken. Since nearly all of the reciprocal data are decimals less than 1, they are multiplied by 10,000, which is convenient for calculation and observation, and the corresponding adjacency matrix $G'_a$ is obtained. Some air transportation distance relationships are listed in Table 4.

**Table 4.** Air transportation distance relationships (Unit: km).

| Distance | | Node Number | | | | | |
|---|---|---|---|---|---|---|---|
| | | 6 | 20 | 21 | 23 | 33 | 35 |
| Node Number | 6 | 0 | 0 | 0 | 947 | 1909 | 0 |
| | 20 | 0 | 0 | 1099 | 1015 | 1910 | 1867 |
| | 21 | 0 | 1099 | 0 | 0 | 1133 | 792 |
| | 23 | 947 | 1015 | 0 | 0 | 1545 | 1102 |
| | 33 | 1909 | 1910 | 1133 | 1545 | 0 | 477 |
| | 35 | 0 | 1867 | 792 | 1102 | 477 | 0 |

By importing $G'_a$ into Ucinet, the weighted network diagram of air transportation is obtained, as shown in Figure 2, which adds distance as the edge weight on the basis of Figure 1.

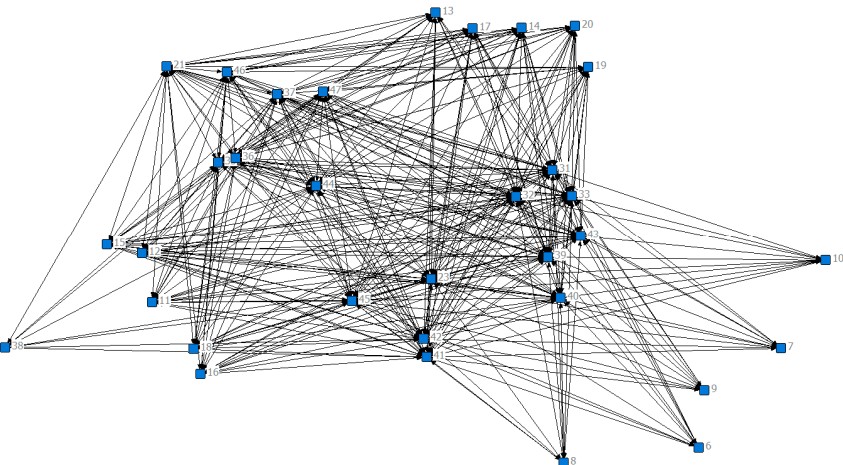

**Figure 2.** Weighted network diagram of air transportation.

Through centrality and weighted measurement analyses, the node strength of the air transportation network is determined to calculate the unit weight of the nodes and the difference of the nodes. The characteristic details are shown in Table 5 (enumeration). In the table, Degree refers to the node

strength of each node, nDegree refers to the relative node strength, Unit Weight refers to the unit weight of each node and Node Difference refers to the weight difference of each node edge.

(3) Analysis of the Node Integrated Air transportation Capability

The above analysis can be used to obtain four indicators: transportation accessibility, node centrality, unit distance between nodes and the difference between nodes in the air transportation network. The original data are standardized, and the values are weighted to determine the comprehensive ability score.

**Table 5.** Node strength, unit weight and node difference of the air transportation weighted network.

| Node Number | Degree ($S_i$) | nDegree | Unit Weight ($U_i$) | Node Difference ($Y_i$) |
|---|---|---|---|---|
| 6 | 61.174 | 0.051 | 6.1174 | 0.10904 |
| 16 | 123.546 | 0.103 | 7.267412 | 0.072937 |
| 22 | 0 | 0 | 0 | 0 |
| 23 | 315.359 | 0.263 | 10.51197 | 0.038922 |
| 31 | 248.807 | 0.208 | 8.885964 | 0.053532 |
| 35 | 257.028 | 0.215 | 10.28112 | 0.05123 |
| 37 | 273.933 | 0.229 | 11.41388 | 0.047883 |
| 38 | 80.944 | 0.068 | 10.118 | 0.141036 |
| 39 | 246.417 | 0.206 | 9.126556 | 0.054719 |
| 40 | 246.417 | 0.206 | 9.126556 | 0.054719 |
| 41 | 284.556 | 0.238 | 9.4852 | 0.039101 |
| 42 | 284.556 | 0.238 | 9.4852 | 0.039101 |
| 43 | 182.136 | 0.152 | 7.005231 | 0.045508 |
| 45 | 124.420 | 0.104 | 4.443571 | 0.040796 |
| 46 | 248.838 | 0.208 | 11.31082 | 0.050837 |
| 47 | 270.548 | 0.226 | 11.76296 | 0.049055 |

The dimensionless data are detailed in Table 6 (retaining the four decimal places and ranking them in descending order according to the comprehensive weight, listing the top ten).

**Table 6.** Structural characteristics and comprehensive weights of the air transportation network.

| Node Number | Degree (K) | Betweenness (B) | Unit Weight (U) | Node Difference (Y) | Comprehensive Weights (T) |
|---|---|---|---|---|---|
| 23 | 1.4007 | 2.0910 | 1.1711 | −0.2254 | 4.4373 |
| 41 | 1.4007 | 2.1903 | 0.9198 | −0.2207 | 4.2901 |
| 42 | 1.4007 | 2.1903 | 0.9198 | −0.2207 | 4.2901 |
| 31 | 1.2208 | 1.6516 | 0.7732 | 0.1596 | 3.8052 |
| 32 | 1.2208 | 1.6516 | 0.7732 | 0.1596 | 3.8052 |
| 33 | 1.2208 | 1.6516 | 0.7732 | 0.1596 | 3.8052 |
| 39 | 1.1309 | 1.4903 | 0.8321 | 0.1908 | 3.6441 |
| 40 | 1.1309 | 1.4903 | 0.8321 | 0.1908 | 3.6441 |
| 43 | 1.0409 | 1.6134 | 0.3130 | −0.0519 | 2.9155 |
| 45 | 1.2208 | 2.0326 | −0.3137 | −0.1760 | 2.7637 |

### 3.2.2. Analysis of the Railway Transportation Network

Most of timber transportation does not use air transportation, except for a few rare varieties; it mainly uses railway, highway and waterway (only three markets including timber port are involved in this paper, which lacks certain practical significance, so waterway transportation is temporarily not considered). According to statistics, the railway conditions of 47 timber markets are shown in Table 7.

(1) Analysis of Railway Transportation Network

Using the network crawler technology, the domestic railway shift situation is obtained, and the relationship between shifts is transformed into the adjacent matrix $G_r$. The matrix diagram is obtained by Ucinet software analysis, and the node degree and betweenness centrality for each node are calculated.

(2) Analysis of Railway Transportation Distance

To ensure the indicator changes in the same direction, the reciprocal of time is taken and transformed into the corresponding adjacency matrix $G_{r'}$. Ucinet is used to calculate the unit weight of nodes and the differences of nodes.

**Table 7.** Railway transportation conditions in 47 timber markets.

| Node Number | Railway Station | Node Number | Railway Station | Node Number | Railway Station | Node Number | Railway Station |
|---|---|---|---|---|---|---|---|
| 1 | Dongguan | 13 | Guangzhou South | 25 | JiaXing | 37 | Changsha West |
| 2 | Dongguan | 14 | Guangzhou East | 26 | Huzhou | 38 | Yueyang East |
| 3 | Dongguan | 15 | Guangzhou West | 27 | Huzhou | 39 | Shijiazhuang North |
| 4 | Dongguan East | 16 | Guangzhou North | 28 | Suzhou North | 40 | Shijiazhuang North |
| 5 | Dongguan | 17 | Guangzhou East | 29 | Suzhou North | 41 | Shanghai West |
| 6 | Foshan | 18 | Guangzhou East | 30 | Zhenjiang | 42 | Shanghai West |
| 7 | Foshan | 19 | Guangzhou | 31 | Tianjin North | 43 | Beijing East |
| 8 | Foshan | 20 | Guangzhou west | 32 | Tianjin | 44 | Mudanjiang |
| 9 | Foshan | 21 | Hangzhou | 33 | Tianjin West | 45 | Manzhouli |
| 10 | Foshan | 22 | Shaoxing | 34 | Dezhou | 46 | Fu-Yang |
| 11 | Guangzhou East | 23 | Wenzhou South | 35 | Qingdao North | 47 | Nam Cheong |
| 12 | GuangZhou East | 24 | JiaXing | 36 | Qingdao North | | |

The detailed data after dimensionless processing are shown in Table 8 (retaining the last four decimal places and listing the top 10 in descending order of comprehensive weight).

**Table 8.** Structural characteristics and comprehensive weights of the railway transportation network.

| Node Number | Degree (K) | Betweenness (B) | Unit Weight (U) | Node Difference (Y) | Comprehensive Weights (T) |
|---|---|---|---|---|---|
| 33 | 1.7773 | 3.2841 | −0.8686 | 0.2919 | 4.4847 |
| 32 | 1.7773 | 3.2841 | −1.0086 | −0.9868 | 3.0661 |
| 31 | 1.7773 | 3.2841 | −0.9997 | −1.0138 | 3.0479 |
| 43 | 1.3467 | 1.7569 | −0.8147 | 0.3273 | 2.6162 |
| 21 | 1.2391 | 0.9607 | −0.4172 | −0.0307 | 1.7518 |
| 42 | 1.0238 | 0.4129 | −0.5016 | 0.5034 | 1.4385 |
| 41 | 1.0238 | 0.4129 | −0.4942 | 0.4558 | 1.3983 |
| 47 | 1.5620 | 1.1924 | −0.9599 | −1.3303 | 0.4643 |
| 15 | 0.0550 | −0.2140 | 0.6177 | 0.0040 | 0.4627 |
| 16 | 0.0550 | −0.2140 | 0.6177 | 0.0040 | 0.4627 |

### 3.2.3. Highway Transportation Network Analysis

For different timber markets in the same city, because the distance is relatively close, highway transportation is used. The highway transportation indicators analyzed are used to measure the advantages and disadvantages of the timber market under the jurisdiction of the same city.

(1) Analysis of the Highway Transportation Network

Using the network crawler technology, the domestic highway shift is obtained, and the relationship between shifts is transformed into the adjacency matrix $G_h$. The matrix diagram is obtained by Ucinet software analysis, as shown in Figure 3. The node degree and betweenness centrality for each node are calculated.

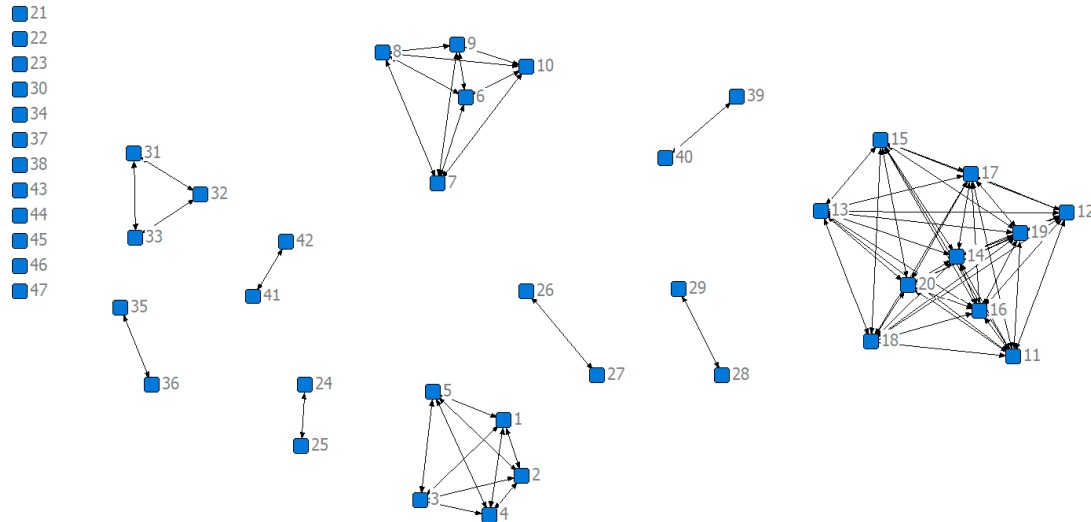

**Figure 3.** Unweighted network diagram of highway transportation.

(2) Analysis of Highway transportation Distance

To ensure the indicator changes in the same direction, the reciprocal distance is taken and transformed into the corresponding adjacency matrix $G_h'$. Ucinet is used to calculate the unit weight of nodes and the difference of nodes.

The measured characteristic data are shown in Table 9 (the top 10 enumeration numbers). Since the betweenness values are all equal to 0, they are not listed. Among them, $K_i$, $S_i$, $U_i$ and $Y_i$ represent the difference of centrality, node strength, unit weight and edge weight, respectively; $K, U, Y$ and $T$ represent centrality, unit weight, edge weight difference and comprehensive weights, respectively, after standardization.

**Table 9.** Structural characteristics and comprehensive weights of the highway network.

| Node Number | $K_i$ | $S_i$ | $U_i$ | $Y_i$ | K | U | Y | Comprehensive Weights (T) |
|---|---|---|---|---|---|---|---|---|
| 1 | 4 | 208.946 | 52.2365 | 0.6715 | 0.2512 | 4.0946 | 0.4939 | 4.8396 |
| 2 | 4 | 204.788 | 51.197 | 0.6936 | 0.2512 | 3.9994 | 0.5433 | 4.7939 |
| 3 | 4 | 66.9 | 16.725 | 0.3726 | 0.2512 | 0.8443 | −0.1737 | 0.9218 |
| 4 | 4 | 23.579 | 5.8948 | 0.2531 | 0.2512 | −0.1469 | −0.4406 | −0.3363 |
| 5 | 4 | 16.733 | 4.1833 | 0.2521 | 0.2512 | −0.3036 | −0.4428 | −0.4951 |
| 6 | 4 | 78.505 | 19.6263 | 0.5407 | 0.2512 | 1.1099 | 0.2018 | 1.5628 |
| 7 | 4 | 32.642 | 8.1605 | 0.4657 | 0.2512 | 0.0605 | 0.0343 | 0.3460 |
| 8 | 4 | 83.414 | 20.8535 | 0.5055 | 0.2512 | 1.2222 | 0.1231 | 1.5966 |
| 9 | 4 | 41.061 | 10.2653 | 0.3746 | 0.2512 | 0.2531 | −0.1692 | 0.3352 |
| 10 | 4 | 32.796 | 8.199 | 0.4624 | 0.2512 | 0.0640 | 0.0269 | 0.3422 |

## 3.3. Analysis of the Logistics Capacity Coefficient Combined with the Freight Volume

It is difficult to obtain data for timber turnover because the information level of the timber market is low. According to research, the market scale and timber turnover of enterprises in each market are similar and the number of enterprises in the market is proportional to the freight volume in a sense, so it is relatively feasible to replace the freight volume of each market with the number of enterprises in the market.

The number of enterprises stationed in each market is shown in Table 10. As an indicator to measure the freight volume in each market, $P$ is the value after standardization of the number of enterprises located in each market.

**Table 10.** Analysis of freight volume.

| Node Number | Number of Enterprises | P | Node Number | Number of Enterprises | P | Node Number | Number of Enterprises | P |
|---|---|---|---|---|---|---|---|---|
| 1 | 346 | 1.95 | 17 | 7 | −0.54 | 33 | 32 | −0.36 |
| 2 | 26 | −0.40 | 18 | 12 | −0.50 | 34 | 136 | 0.41 |
| 3 | 771 | 5.07 | 19 | 5 | −0.55 | 35 | 37 | −0.32 |
| 4 | 292 | 1.55 | 20 | 9 | −0.52 | 36 | 44 | −0.27 |
| 5 | 6 | −0.55 | 21 | 59 | −0.16 | 37 | 102 | 0.16 |
| 6 | 66 | −0.11 | 22 | 10 | −0.52 | 38 | 8 | −0.53 |
| 7 | 69 | −0.08 | 23 | 20 | −0.44 | 39 | 44 | −0.27 |
| 8 | 54 | −0.19 | 24 | 29 | −0.38 | 40 | 37 | −0.32 |
| 9 | 10 | −0.52 | 25 | 14 | −0.49 | 41 | 90 | 0.07 |
| 10 | 8 | −0.53 | 26 | 33 | −0.35 | 42 | 384 | 2.23 |
| 11 | 4 | −0.56 | 27 | 24 | −0.41 | 43 | 116 | 0.26 |
| 12 | 10 | −0.52 | 28 | 224 | 1.05 | 44 | 140 | 0.44 |
| 13 | 11 | −0.51 | 29 | 33 | −0.35 | 45 | 83 | 0.02 |
| 14 | 228 | 1.08 | 30 | 30 | −0.37 | 46 | 31 | −0.36 |
| 15 | 10 | −0.52 | 31 | 35 | −0.33 | 47 | 8 | −0.53 |
| 16 | 7 | −0.54 | 32 | 30 | −0.37 | | | |

The comprehensive weights of the three modes of transportation are obtained through the above calculation and analysis. Weighted for railway and air transportation, according to the data from the official website of the National Bureau of Statistics in 2018, the railway freight volume is 4.03 billion tons and the air freight volume is 7.39 million tons, i.e., railway transportation accounts for 99.82% and air transportation accounts for 0.18%. Therefore, the comprehensive weights of air and railway transportation combined with the comprehensive weight of highway transportation are used to analyze the comprehensive transportation indicators of each timber market. It is reflected by the transportation capability coefficient.

The final determination of the logistics capacity coefficient is obtained by weighting the transportation capacity indicator and the freight volume indicator. Logistics capacity coefficients of different timber centers in the same city, which are obtained according to the ranking of highway comprehensive weights, are shown in Table 11 (listing the top ten).

**Table 11.** Calculation and sorting of the logistics capacity coefficient.

| Node Number | Comprehensive Weight of Air Transportation | Comprehensive Weight of Railway Transportation | Comprehensive Weight of Highway Transportation | Transportation Capability Coefficient | Comprehensive Weight of Freight Volume | Logistics Capacity Coefficient |
|---|---|---|---|---|---|---|
| 1 | −4.6581 | 0.4279 | 4.8396 | 5.2583 | 1.9480 | 7.2062 |
| 3 | −4.6581 | 0.4279 | 0.9218 | 1.3405 | 5.0663 | 6.4068 |
| 2 | −4.6581 | 0.4279 | 4.7939 | 5.2125 | −0.4000 | 4.8125 |
| 42 | 4.2901 | 1.4385 | 0.4041 | 1.8479 | 2.2268 | 4.0747 |
| 33 | 3.8052 | 4.4847 | −0.1875 | 4.2960 | −0.3559 | 3.9401 |
| 14 | 0.7542 | 0.4596 | 1.1079 | 1.5681 | 1.0822 | 2.6503 |
| 31 | 3.8052 | 3.0479 | −0.2881 | 2.7612 | −0.3339 | 2.4273 |
| 18 | 0.7542 | 0.4596 | 2.3145 | 2.7747 | −0.5027 | 2.2720 |
| 12 | 0.7542 | 0.4596 | 2.3201 | 2.7802 | −0.5174 | 2.2629 |
| 32 | 3.8052 | 3.0661 | −0.6708 | 2.3966 | −0.3706 | 2.0260 |

A scatter plot of the logistics capacity coefficient of each node is shown in Figure 4.

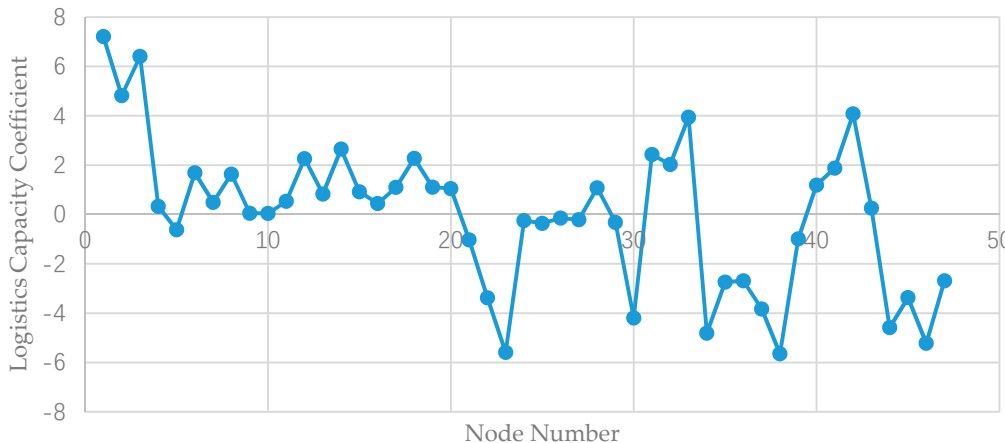

**Figure 4.** Logistics capacity coefficient of each node.

*3.4. Grade Setting of the Timber Logistics Center*

Node is the 47 timber trading markets discussed in this paper, covering 22 cities. The number of prefecture-level or municipality directly under the central government and the number of each market are shown in Table 12.

**Table 12.** Correspondence between city number and node number.

| Node Number | City Name | City Number | Node Number | City Name | City Number |
|---|---|---|---|---|---|
| 1 | Dongguan | 1 | 11 | Guangzhou | 13 |
| 2 | Dongguan | 1 | 12 | Guangzhou | 13 |
| 3 | Dongguan | 1 | 13 | Guangzhou | 13 |
| 4 | Dongguan | 1 | 14 | Guangzhou | 13 |
| 5 | Dongguan | 1 | 15 | Guangzhou | 13 |
| 6 | Foshan | 2 | 16 | Guangzhou | 13 |
| 7 | Foshan | 2 | 17 | Guangzhou | 13 |
| 8 | Foshan | 2 | 18 | Guangzhou | 13 |
| 9 | Foshan | 2 | 19 | Guangzhou | 13 |
| 10 | Foshan | 2 | 20 | Guangzhou | 13 |
| 21 | Hangzhou | 3 | 31 | Tianjin | 14 |
| 22 | Shaoxing | 4 | 32 | Tianjin | 14 |
| 23 | Wenzhou | 5 | 33 | Tianjin | 14 |
| 24 | Jiaxing | 6 | 34 | Dezhou | 15 |
| 25 | Jiaxing | 6 | 35 | Qingdao | 16 |
| 26 | Huzhou | 7 | 36 | Qingdao | 16 |
| 27 | Huzhou | 7 | 37 | Changsha | 17 |
| 28 | Suzhou | 8 | 38 | Yueyang | 18 |
| 29 | Suzhou | 8 | 39 | Shijiazhuang | 19 |
| 30 | Zhenjiang | 9 | 40 | Shijiazhuang | 19 |
| 41 | Shanghai | 10 | 45 | Manzhouli | 20 |
| 42 | Shanghai | 10 | 46 | Fuyang | 21 |
| 43 | Beijing | 11 | 47 | Nanchang | 22 |
| 44 | Mudanjiang | 12 | | | |

The logistics capacity is measured for each city. The logistics capacity coefficient is the average value of the logistics capacity coefficients of all timber trading markets in the city. The data are shown in Table 13 below.

The rules for network construction are formulated. The first 10% of cities have the highest grade, the second 20% of cities have the second grade, the third 30% of cities have the third grade and the last 40% of cities have the fourth grade. The nodes with the highest score are selected as logistics centers in each city.

**Table 13.** Grade setting of the city logistics center.

| City Number | Grade of Logistics Center | City Name | Logistics Capacity Coefficient |
|---|---|---|---|
| 1 | Level 1 | Dongguan | 3.625992 |
| 10 | | Shanghai | 2.976004 |
| 14 | Level 2 | Tianjin | 2.797807 |
| 13 | | Guangzhou | 1.315718 |
| 2 | | Foshan | 0.779902 |
| 8 | | Suzhou | 0.381721 |
| 11 | | Beijing | 0.255385 |
| 19 | Level 3 | Shijiazhuang | 0.100643 |
| 7 | | Huzhou | −0.18127 |
| 6 | | Jiaxing | −0.30508 |
| 3 | | Hangzhou | −1.02727 |
| 22 | | Nanchang | −2.68646 |
| 16 | | Qingdao | −2.71568 |
| 20 | | Manzhouli | −3.37272 |
| 4 | Level 4 | Shaoxing | −3.3756 |
| 17 | | Changsha | −3.83059 |
| 9 | | Zhenjiang | −4.19341 |
| 12 | | Mudanjiang | −4.58568 |
| 15 | | Dezhou | −4.8129 |
| 21 | | Fuyang | −5.22204 |
| 5 | | Wenzhou | −5.5914 |
| 18 | | Yueyang | −5.65007 |

According to Table 13, Dongguan and Shanghai are established as first-level logistics center cities. According to the logistics capacity coefficients of different nodes in the city, No. 1 node, Dongguan Houjie·Xingye Timber Market, is set as the timber logistics center of Dongguan City and the No. 42 node, Shanghai Baoshan Furen Timber Market, is set as the timber logistics center of Shanghai. Similarly, Tianjin, Guangzhou, Foshan, Suzhou and Beijing are established as secondary logistics center cities. The logistics centers are: 33 Tianjin Beichen District Hanjiashu Timber Market, 14 Guangdong Guangzhou Yuzhu International Timber Market, 6 Foshan Shunde ShuiTeng Timber Market, 28 Zhangjiagang Timber Exchange Market and 43 Beijing Dongba Imported Timber Market. Then, Shijiazhuang, Huzhou, Jiaxing, Hangzhou, Nanchang, Qingdao and Hulunbuir are established as third-tier logistics center cities, whose logistics centers are 40 Shijiazhuang City Timber Trading Market, 26 Huzhou Nanxun Building Materials Market, 24 Jiaxing Jiashan International Timber Market, 21 Hangzhou Timber Trading Market and 47 Jiangxi Jinxian Wengang Wild Chicken Cage Timber Market, 36 Qingdao Licang District Wantou First Timber Market and 45 Inner Mongolia Manzhouli City Timber Market. Finally, Shaoxing, Changsha, Zhenjiang, Mudanjiang, Dezhou, Fuyang, Wenzhou and Yueyang are established as fourth-tier logistics center cities, and their logistics centers are 22 Shaoxing Tashan Timber Market and 37 Hunan Changsha Zhongnan Timber Market, 30 Zhenjiang Yangzhong Venus Timber Market, 44 Heilongjiang Suifenhe Imported Timber market, 34 Shandong Dezhou Timber Market, 46 Anhui Jieshou Lvzhai Timber Market, 23 Wenzhou Timber Exchange Market and 38 Yueyang City Sun Bridge Building Materials Market.

This paper constructs a timber logistics network with Dongguan and Shanghai as the first-level logistics centers; Tianjin, Guangzhou, Foshan, Suzhou, Beijing as the second-level logistics centers, Shijiazhuang; Huzhou as the third level logistics center and the remaining eight cities as the fourth level logistics centers.

The functions of the logistics center mainly include concentrating freight, distribution processing, circulation processing, transportation, storage, transfer, information service, etc. The first-level logistics centers should have all these functions, while the second-level logistics centers contain at least four functions, the third level logistics center has more than two functions and the fourth

level logistics centers contain at least one function. Through the organic connection between the logistics center and the corresponding timber market, the circulation speed of timber logistics can be accelerated, the circulation time can be shortened and the circulation cost can be reduced. In addition, according to the needs of the appropriate processing, rational use of the source of goods can improve economic efficiency.

As the first and second logistics centers, seven cities have high logistics capacity coefficients. These cities are mainly concentrated in the Yangtze River Delta, Pearl River Delta and Bohai rim, among which Shanghai, Dongguan and Suzhou are all members of China's superior timber market. These areas are the regions undergoing rapid development of the timber processing industry in China. Their economic development is very active, and private and foreign enterprises are concentrated in these regions. These areas are the first choice for foreign enterprises to set up factories in China. The region has a strong economic strength and a good economic foundation. The total production and consumption of timber processing and its products are among the best in the country. This region is an important production base for wood-based panels as well as wood and wood bamboo flooring products in China. These areas easily form industrial clusters, which is helpful to promote the growth of the regional economy. In addition, more than 55% of China's timber consumption is dependent on foreign countries. These cities are located in the main port for imported timber and the areas surrounding the port. Imported timber forms a large timber trading market, log processing and timber logistics gathering area nearby. In the future, advanced technologies such as big data can be used to reasonably analyze the fluctuation of the domestic market, which also can adopt the operation mode of cross-border e-commerce according to market demand. On one hand, it can reduce the trade cycle of timber purchasing and transportation. On the other hand, it can reduce the storage cost according to market demand. It is of great significance to set up timber logistics centers in these locations, which can play an effective role in linking the market together.

As the third level logistics centers, seven cities are distributed in Hebei, Zhejiang, Jiangxi, Shandong, Neimenggu and other regions. The timber processing industry in these regions is relatively developed. As the main cities in Eastern China, the timber processing industry shows strong development potential. There is a certain distance from the main timber import port; it can be used as a powerful supplement to the first and second logistics centers to promote the development of the timber market.

In addition, as transitional cities between the East and the West, eight level four logistics centers have begun to transfer the manufacturing industry to these regions in recent years. Due to the rising comprehensive costs of land, labor and raw materials, environmental protection requirements, such as industrial transfer, can not only promote the local economic development but also facilitate the use of processing and manufacturing space in the central and western regions. All of these make this area have the advantages of low labor costs and close-to-plantation resources, which is a powerful guarantee for the development of the timber industry.

## 4. Comparative Analysis with Other Methods

In this paper, the complex network method is used to analyze and weight three kinds of timber transportation modes, so that we can get the transportation capability coefficient, and know the initial location of the timber logistics center. Then, with the comprehensive weighted analysis of freight volume, we can get the method of setting the level of logistics center. The main methods of the logistics center location and the application research results of complex network method in location selection are compared with the previous research methods, which are shown in Table 14 below:

**Table 14.** Comparative analysis with previous methods.

| Method | Brief Description | Advantages | Disadvantages | Applicability |
|---|---|---|---|---|
| With subjective weight assignment | All of these methods are based on the combination of different mathematical models and subjective evaluation indicator system, so as to achieve the location of logistics center. | These methods consider specific objectives, such as the lowest cost or the closest distance, and the assignment of subjective evaluation indicators. | It is difficult to construct the evaluation indicator system comprehensively. The quantization method is not easy to obtain. | This kind of method is suitable for long-term planning and in a small region. |
| No subjective weight assignment | These methods make use of different mathematical models and algorithms to study location quantitatively. | These methods mainly consider economic benefit and time benefit. | Customer needs and costs will vary with time and location. | This kind of method is suitable for industries with small market changes, otherwise dynamic planning should be used as an assistant. |
| Subjective weighting of complex networks | This kind of method is based on complex network method and subjective evaluation indicator. | According to the complex network characteristics of each node, it is more objective. | The subjective evaluation indicator is difficult to obtain and the accuracy needs to be improved. | It is suitable for the node selection of long-term planning with small area, which constitute the nodes of complex network. |
| Objective weighting of complex networks (method used in this paper) | The method of this paper is to use the complex network method, combined with the comprehensive weight of freight volume, which can determine the important nodes and grade settings. | According to the characteristics of nodes and the weighting of main objective data, the conclusion is not interfered by subjective factors. | The weighting method can be improved. | This method is suitable for the initial planning of the project, the data of the research object is difficult to obtain, the research object is special, and there are alternative positions in a large area. |

Compared with previous studies, this method avoids the location results caused by subjective qualitative analysis or quantitative calculation, which may be unreasonable in practice. On the basis of the existing position of the timber trading market, this paper uses the complex network theory to evaluate the importance of transportation network nodes, which is of great practical significance. The method can greatly reduce the cost and risk of initial research when selecting the location of the logistics center, and the location is close to the timber demand market. In this way, we can use a reasonable transportation mode to carry out logistics transportation and other related businesses. Due to the particularity of timber itself, the location of timber supply is concentrated in some regions of the country, but the demand location is relatively scattered. Therefore, we need to make full use of the role of different levels of logistics centers in the regional logistics network. In addition, with the help of modern technology such as big data, it can be applied to the node characteristics of complex networks, which makes data analysis more reliable. For the first- and second-level logistics centers, we should increase investment and focus on management, so that their services can reach the third and fourth level logistics centers as much as possible, and gradually form a point axis system to promote the development of regional and even national timber logistics.

## 5. Conclusions

This paper put forward the analysis model and the method of the timber transportation network based on 47 timber trading markets in China, from data acquisition, data processing and data visualization, to analyze transportation network characteristics and establish indicators based on the complex network. Through comprehensive consideration of the transportation network and freight volume, the logistics capacity coefficient of each node is set, and finally, the logistics transportation

network center node is determined to realize the grade setting of the logistics center, which provides a new idea for the location of the timber logistics center.

The model and analysis methods proposed in this paper are not only suitable for the reasonable layout of the transportation network and logistics center nodes of timber logistics enterprises but also helpful for the government to consider timber logistics policies, which can save resources and promote the sustainable development of society. Of course, in practice, these analyses should be combined with the specific geographical location of the timber trading market, changes in the surrounding economic environment, national and local government policies and macro development planning to establish the level and scale of the timber logistics center.

In practice, due to the lack of data in a certain field or the difficulty of statistics, it is not easy to obtain theoretical verification of the centrality and grade evaluation of the relevant transportation network nodes. With the help of the connectivity of each transportation mode between nodes in the existing transportation network, as well as the actual or equivalent transportation volume of the corresponding market between nodes, reasonable explorations and assumptions can be performed comprehensively. In this way, useful references for the transportation network of people or freight in the same city and inter-city can be provided in terms of finding the central node and corresponding level setting.

In addition, for the selection of indicators in transportation network analysis, the weighting methods of railway and air transportation and the selection of timber transportation nodes, we hope to have more appropriate methods to obtain relevant information in the future.

**Author Contributions:** X.H. is responsible for data collection and co-ordination. L.X. provided an interpretation of the results and wrote the majority of the paper. Y.W. collects data and completes the calculation and collation of some data. X.Y. contributed to the paper review and editing. Y.Z. contributed to the paper review and editing. All authors have read and agreed to the published version of the manuscript.

**Funding:** This research was funded by National Natural Science Foundation of China (grant no. 71701099) and Basic Research Program of Science and Technology Commission Foundation of Jiangsu Province (grant no. BK20180775).

**Acknowledgments:** The authors would like to express their sincere thanks to the anonymous reviewers for their constructive comments on an earlier version of this manuscript.

**Conflicts of Interest:** The authors declare no conflicts of interest.

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
