# Peer review of "Grade Setting of a Timber Logistics Center Based on a Complex Network: A Case Study of 47 Timber Trading Markets in China"

_information, doi:10.3390/info11020107_

Round 1

Reviewer 1 Report

All the remarks presented i my first review have been included. I have not furhter remarks to the authors.

Reviewer 2 Report

Thank you.

Reviewer 3 Report

This paper analyzes the planning of timber logistics centers in China based on 47 timber trading market information.  As for the technical aspect, the authors have used centrality measures based on graph network theory.  The comments are provided below.

It would not be accurate to provided logistics planning decisions just based centrality measures which are based on the distances in different modes.  Without considering any specific objective and associated constraints, it is not clear how realistic the decisions would be.

The writing needs significant improvement.  There are hundreds of grammatical, proof-reading and consistency errors.

The findings/discussions are compared against existing studies and/or information.  Without that, how do we know the results obtained from the model presented are useful.

Reviewer 4 Report

The authors have applied network theory concepts to set the grade of city logistics centres of timber markets in China. The network model includes the computation of comprehensive weights for air, railway and highway transportation for each logistics centre. In addition, the freight volume has been analysed in order to obtain logistics capacity coefficient.

Although the paper seems to be mathematically sound, there are some issues that could be addressed:

My main concern is about the contribution to the literature of the paper. Apart from the relevance of the application to the timber logistics in China, the authors should point out which advantages their methodology provides over previous approaches. In addition, the relation with previous approaches should be explained. In my opinion, the number of tables in section 3 could be reduced. I encourage the authors to try to synthesize data from tables 9 to 12 into a fewer number of tables. Could the authors highlight some managerial insight derived from their study?

Round 2

Reviewer 3 Report

The reviewer is not completely clear on the response to the first comment.  What specific criteria were considered? What improvement/addition was done to the manuscript? The authors need to do a better job of explaining.  

Comment #3: A new table has been added to provide some details on previous methods relevant to the study.  However, there exists no reference for those methods.  Also, it is not just a matter of advantages or disadvantages across the studies. It is required to explain the novelty of the proposed approach. 

Reviewer 4 Report

The authors have sucessfully addressed all my comments and suggestions. In my opinion, the paper is ready to be published.

This manuscript is a resubmission of an earlier submission. The following is a list of the peer review reports and author responses from that submission.

Round 1

Reviewer 1 Report

The authors of the article “Analysis and optimization of Timber transportation network based on complex network: A case study on 47 timber trading centres in China”, basing on the methods of complex network theory and data mining, analysed the importance of timber trading centres in China. Finally, they divided the analysed timber centres into four groups: first, second, third and fourth-class transport cities, stating that the first-level logistics nodes can have all functions (centralized cargo, distribution processing, circulation processing, storage, transit, information service), while the second-level logistics nodes contain four functions, the third-level nodes have two or more, and the fourth-level only one function. (line 410-414). In the conclusion the Authors stated that they presented a new idea for the optimization of wood material transportation network.

The article needs to be improved.

First of all, no clear aim of the article is presented. The authors clearly presented what they did, but they didn’t present what it was for and what was the practical value of the research. The aim of the research should be clearly described in the Abstract and in the Introduction. If the aim of the article was to present the new method of classification of the logistics centers, there should be a validation of it (e.g. Authors should check if the analysed logistics centers perform the functions according to the class-levels to which they were assigned). I am not also convinced if the assumptions were correctly adopted. The Authors determined the freight volume by the number of enterprises stationed in each market. The timber monthly/yearly turnover would better determine the freight volume. Likewise, transport cost, rather than distance or delivery time, would better determine the “Complex Traffic Network Model” (line 201-204). There are also some mistakes in the calculation of percentage of the freight volume (line 353-354).

Despite above remarks, the concept proposed by the authors seems to be developmental and worth presenting to the readers of the Information.

Reviewer 2 Report

The paper studied the optimization of the timber transportation network in China. The paper employed complex network theory to solve the optimal trading centre location problem. The case study used the data from 47 trading centre in China. The analysis results are based on three different transportation network. The follows are the comments to the authors:

The title of the paper suggests the optimization and complex network theory, however, when I read into the method, there is not much of optimization method and complex network theory in the paper. In section 2.1 and 2.2, the description of the basic network problem is not very rigorous. In common graph network, V stands for vertices. ‘node betweenness’? That should be ‘betweenness centrality for node’. Also, the vector should be bold and italic. Technically transportation network and traffic network are two different thing. In most of places, authors use the traffic network instead of transportation network. Section 2.3 is very vaguely written, what are the optimal modelling method? What are the constraints for each nodes? What complex network method is used here? It’s all not very clear to me. Section 2.4, the title is ‘transportation network optimisation model’, this is a very technical term for people working the transportation field, what are the formulation of the model? What are your objectives and constraints? What optimization method you employed? The authors didn’t explain anything here. The authors also mentioned the word ‘big data’, what exactly the ‘big data’ dimension of this paper? The authors did not even mention the amount of data and complexity involved in this data mining process. It’s not very scientific at all. Is this a directed graph or undirected graph? The authors did not seem to mention in the paper. The concept of ‘traffic capacity’ index is a bit confusing. This is essentially a proxy of transportation accessibility and strength of each node in the network. It’s not the same concept as people would use in the transportation field, it normally means the overall capacity of a node. It needs to be clarified or changes to another term. The paper is difficult to follow in many places, i would strongly suggest going through a proof-read before the submission. Some other minor mistakes: Be consistent with the spelling of ‘transportation’ and ‘transport’. Variables in the paper should be italic.

Reviewer 3 Report

The presented research work contains a number of methodological and editorial incorrections:

1) The title indicates optimization and the work contains only analysis. There is no identification of the research problem related to the examined process (assessment of needs of timber transportation market).

2) The literature review is quantitative and does not justify the need for research.

3) Choosing a timber transport market to test the presented methodology seems inappropriate. This is a very specific market with unique transport needs as evidenced by its modal structure of transport.

4) The conclusions regarding the assignment of functions of logistics transport nodes have no confirmation in the research.

5) It is incomprehensible to identify the number of enterprises in the node with the concept of fright volume (table 11). 

6) The relationship between Logistics Capability and Logistics value is incomprehensible (table 12 and 13).

7) It is not explained why in most tables values ​​of coefficients are given only for some nodes (node ​​number). The adopted numbering of cities (city number) and its relation to numbering of nodes is unclear.

8) Logical mistake in the sentence: "... railway freight volume is 4.03 million tons and air freight volume is 7.39 million tons, i.e., aviation accounts for 0.18% and railway station accounts for 99.82%".

9) Some names adopted for the description of transport processes are incorrect: 'the longest node in the shortest path', 'the valley time', 'highway transport'.

Round 2

Reviewer 3 Report

Thanks for your corrections. Title and content are much more consistent. I see still need for language correction. 

I will appreciate conclusions regarding the wider use of the method proposed, i.e. with regard to other transport systems.